# Clonal Evolution of Multiple Myeloma—Clinical and Diagnostic Implications

**DOI:** 10.3390/diagnostics11091534

**Published:** 2021-08-25

**Authors:** Aleksander Salomon-Perzyński, Krzysztof Jamroziak, Eliza Głodkowska-Mrówka

**Affiliations:** 1Department of Hematology, Institute of Hematology and Transfusion Medicine, 14 I. Gandhi St., 02-776 Warsaw, Poland; asalomon@ihit.waw.pl; 2Department of Hematology, Transplantation and Internal Medicine, Medical University of Warsaw, 1A Banach St., 02-097 Warsaw, Poland; k.m.jamroziak@gmail.com; 3Department of Hematological and Transfusion Immunology, Institute of Hematology and Transfusion Medicine, 14 I. Gandhi St., 02-776 Warsaw, Poland; 4Department of Laboratory Diagnostics and Clinical Immunology of Developmental Age, Medical University of Warsaw, 63 Zwirki i Wigury St., 02-097 Warsaw, Poland; 5Department of Experimental Hematology, Institute of Hematology and Transfusion Medicine, 14 I. Gandhi St., 02-776 Warsaw, Poland

**Keywords:** multiple myeloma, clonal evolution, tumor heterogeneity, genetic heterogeneity

## Abstract

Plasma cell dyscrasias are a heterogeneous group of diseases characterized by the expansion of bone marrow plasma cells. Malignant transformation of plasma cells depends on the continuity of events resulting in a sequence of well-defined disease stages, from monoclonal gammopathy of undetermined significance (MGUS) through smoldering myeloma (SMM) to symptomatic multiple myeloma (MM). Evolution of a pre-malignant cell into a malignant cell, as well as further tumor progression, dissemination, and relapse, require development of multiple driver lesions conferring selective advantage of the dominant clone and allowing subsequent evolution under selective pressure of microenvironment and treatment. This process of natural selection facilitates tumor plasticity leading to the formation of genetically complex and heterogenous tumors that are notoriously difficult to treat. Better understanding of the mechanisms underlying tumor evolution in MM and identification of lesions driving the evolution from the premalignant clone is therefore a key to development of effective treatment and long-term disease control. Here, we review recent advances in clonal evolution patterns and genomic landscape dynamics of MM, focusing on their clinical implications.

## 1. Introduction

Next-generation DNA sequencing (NGS) techniques have provided a broad insight into the genomics of cancer, contributing to our current view of malignancies as continuous evolutionary processes. With the rapid technological advances of NGS, including single cell sequencing techniques, the reduction of sequencing costs, and the optimization of the bioinformatics software, the data provided by NGS are increasingly translated into the clinic. Currently, NGS enriches diagnostic algorithms for cancers, enabling rapid identification of molecular prognostic and predictive factors, as well as markers to track residual disease or target points for cancer therapy. Genetic data navigate the clinician in the decision-making process regarding the choice of treatment for an individual patient. While the contribution of NGS to deciphering cancer biology is fundamental, a key challenge for NGS in clinical oncology is to develop truly personalized cancer therapy, responding appropriately to ongoing changes in the molecular landscape of cancer.

Approximately 33,000 new cases and 20,000 deaths due to multiple myeloma (MM) are reported each year in the European Union [1]. Although novel, highly active agents, such as proteasome inhibitors (PIs) [2,3,4], immunomodulatory drugs (IMiDs) [5,6,7], monoclonal antibodies [8,9,10], together with the high-dose melphalan conditioning regimen followed by autologous stem cell transplantation (autoHSCT) [11] have undeniably improved the MM patients prognosis, the emergence of drug resistance still remains a major treatment complication. Often, it leads to adverse outcomes, and most patients die of relapsed disease.

More and more data suggest that tumor progression, dissemination, and relapse in MM is driven by clonal evolution, a process of natural selection facilitating tumor plasticity and the ability to adapt to the environment leading to formation of genetically complex and heterogenous tumors. NGS techniques recently confirmed that MM is a composition of genetically distinct subclones that evolve through space [12] and time [13,14,15,16,17,18,19,20] following different patterns of clonal evolution. As selection of drug-resistant subclones is thought to contribute to treatment failure in MM [13,14,18], investigation of the processes underlying MM clonal evolution using NGS techniques is particularly important. Here, we briefly present the current state of knowledge about the clonal evolution of MM and discuss its clinical and diagnostic implications.

## 2. Natural History of Disease Evolution in Multiple Myeloma

Symptomatic MM evolves from the preinvasive stage, i.e., monoclonal gammopathy of undetermined significance (MGUS) [21]. Although MGUS is considered to precede nearly all cases of symptomatic MM [22], only 10% of patients with symptomatic MM were previously diagnosed with MGUS, as the course of MGUS is mostly asymptomatic [23]. The annual risk of progression from MGUS to symptomatic MM is 1% [24]. Recently, however, it was found that patients with non-IgM MGUS, with abnormal serum free light chain kappa to lambda ratio (rFLC), and monoclonal protein concentrations exceeding 15 g/L have a higher, 1.8% annual risk of progression to symptomatic MM [25].

Smoldering multiple myeloma (SMM) is an intermediate stage in the evolution of MGUS to symptomatic MM characterized by higher disease burden than MGUS [21]. In the light of the current diagnostic criteria [26], SMM represents a biologically heterogeneous group of plasma cell neoplasms, with variable risk of progression to symptomatic MM [27]. Some authors suggest that this heterogeneity might be explained by various SMM stages: early MGUS-like cases, a slowly progressing subtype with relatively low mutation burden requiring more time for additional genetic events for progression, and more advanced MM-like cases, rapidly progressing and carrying most genetic lesions typically observed in MM [28]. Indeed, the analysis of paired samples obtained before and after progression of SMM to MM showed that high-risk SMM does not differ genetically from MM and therefore requires only time to meet clinical diagnostic criteria of symptomatic MM [29].

Acquisition of the ability of plasma cells to proliferate outside the bone marrow microenvironment determines further evolutionary stages of MM. These include extramedullary MM (EMM), involving the central nervous system, soft tissues, lymph nodes, internal organs and body cavities, and plasma cell leukemia (PCL) characterized by peripheral blood involvement [30].

## 3. Evolutionary Models in Multiple Myeloma

Distinct disease development stages, from MGUS through SMM to symptomatic MM, suggest that malignant transformation of plasma cells depends on the continuity of events ultimately leading to the acquisition of the malignant phenotype (Figure 1). As neoplasms are considered an ecosystem of evolving clones, the basic laws of Darwin’s theory of evolution apply at the cellular level, shaping the phenotype of the developing tumor. Evolution, as a general biological process, can be described as acquisition of heritable changes over successive generations in any population. At the cellular level, limitless replicative potential and excessive proliferation of tumor cells result in relatively rapid formation of multiple generations of genetically distinct subpopulations of cells (subclones) allowing to observe evolutionary processes at the tumor level over a lifetime of a single patient. Thus, tumor evolution begins already at initiation of neoplastic transformation of a single cell in the healthy tissue and further progresses with a growing tumor, formation of new clonal lineages and distinct subpopulations, finally resulting in intratumor heterogeneity.

Oncogenesis is initiated by primary genetic lesions that are clonal in nature, i.e., occur first in the ancestral clone and are “inherited” by all subsequent subclones. During tumor evolution, emerging subclones develop secondary genomic lesions that are consequently present only in a certain fraction of tumor cells, allowing to distinguish the subclones from one another at the genetic level [31]. The dynamics of acquiring new genetic lesions remains heterogeneous. New mutations can be acquired gradually (microevolution) or in a stepwise manner (macroevolution) as a result of massive mutation events, e.g., chromoplexy [14] and chromothripsis [15,17,19].

Acquisition of new driver mutations at the subclonal level is crucial for the development and progression of cancer as it often increases fitness and adaptive potential of the subclone resulting in selective advantage over the remaining subclones, favoring its expansion [32]. However, as mutagenesis is a random process [33], not all the acquired mutations have a positive effect on the competitive abilities of emerging subclones. They may also exert either neutral (passenger mutations) or unfavorable effects, which, under the selective pressure, may result in the elimination of certain subclones [34]. Therefore, although genomic instability may be beneficial to subclones as it may increase their adaptive/competitive potential, after exceeding a certain critical level, it may favor the acquisition of catastrophic genetic lesions, resulting in direct (e.g., by induction of apoptosis) or indirect (by depriving selection advantage) elimination of a certain subclone from the population of cancer cells [34,35]. In this context, clonal evolution results from constant competition of genetically diverse subclonal populations under selective pressure [36]. Similarly to natural selection in ecosystems, selective pressure shapes the subclonal architecture of cancer. The subclones showing the greatest ability to survive in the given microenvironmental conditions are subject to further expansion and begin to dominate the population of cancer cells, while those that have lost the competition are eliminated from the population [36].

Competition between subclones resulting from selective pressure can be described in several theoretical models of evolution dynamics (Figure 2). In the branching evolution model commonly described for most of human neoplasms, tumor subclonal architecture is shaped by clonal competition for limited resources of the microenvironment. Some of the subclones are subject to positive selection and further expansion, giving rise to new progeny populations, while other, poorly adapted subclones, are eliminated from the pool of tumor cells. Therefore, the genetic heterogeneity in branching evolution undergoes constant fluctuations [31,33,35,37]. Independent subclones often undergo parallel evolution by acquiring mutations in the same driver genes that yield equivalent phenotypes. Consequently, competition between similarly fit subclones is limited and several populations may coexist in the tumor mass [13,38].

In the linear evolution model, a single highly adapted subclone acquires such a strong selection advantage that it eliminates all other subclones from the tumor. Consequently, the evolution proceeds sequentially and a single subclonal population is completely replaced by another highly adapted subclone [37]. Therefore, this model represents a snapshot of a specific situation, when only a single clone survives under selective pressure (subclonal sweeps). This evolutionary pattern has been described in acute myeloid leukemia [39], MM [13,20,40], and clear cell renal cancer [41].

Neutral evolution occurs when selection pressure does not play a significant role in shaping the subclonal architecture of cancer [35,37]. This happens when all descendant subclones show a similar ability to survive under certain conditions, i.e., when mutations at the subclonal level do not provide any selective advantage. Due to the lack of selection, the subclonal landscape of the neoplasm is shaped by the genetic drift—random elimination of the existing subclones and the formation of new subclonal populations [33,42]. Williams et al. estimated that this evolutionary pattern may affect approximately 1/3 of malignant tumors [43]. However, this model used to identify neutral evolution was criticized, and the conclusions were questioned [44,45,46,47].

Animal studies of tumor evolution suggested that genetically diverse subclones show a tendency not only to competition, but also to clonal cooperation and interaction. In a murine model of breast cancer, a small subclone population too indolent to compete with other subclones can induce changes in the tumor microenvironment to promote tumor growth. The elimination of this subpopulation by the dominant subclone is in turn associated with tumor regression [48]. Similar conclusions were drawn from a glioblastoma model suggesting that a small subclonal population can maintain the genetic heterogeneity of the tumor, stimulate its growth, and promote local and distant invasion [49]. Moreover, a murine model of MM showed that the inability of the smaller subclone to effectively compete with the dominant subclone does not necessarily lead to its elimination as the dominant subclone may coexist or even promote proliferation of the smaller subclone [50]. Similarly, in other basic research, clonal cooperation has been shown to play an important role both in tumor formation and the acquisition of the invasive phenotype [51,52].

## 4. Genetics of the Evolution of Plasma Cell Dyscrasias

The major factor determining positive selection during evolution is the subclone phenotype, not genotype [53]. The correlation between the genotype and the phenotype of neoplastic cells is not always straightforward [54,55] as the phenotype is also shaped by epigenetics, posttranscriptional RNA modifications, and posttranslational protein modifications [56,57]. Comparison of the results of RNA sequencing and exome sequencing in MM showed that only a small percentage of the mutations are transcribed (23.6–27% of mutations/sample on average), a phenomenon that significantly reduces the mutation pool determining the phenotype of neoplastic cells [58,59]. Despite this, clonal evolution studies focus on genotypes as they enable the definition and differentiation of specific subclonal populations, allowing for the tracking of structural variability of the tumor over time [31].

Genomic landscape of asymptomatic plasma cell dyscrasias as well as the spectrum of genetic changes that occur during progression from MGUS/SMM to symptomatic MM has been recently described [29,60,61,62] (Table 1). Whole-genome sequencing (WGS) of paired samples showed that the genomic structure of SMM is similar to symptomatic MM, and typical MM driver events are observed already at the SMM stage, i.e., t(4; 14), t(11; 14), del(1p), amp(1q21), or mutations in the *NRAS* and *DIS3* genes [61]. Branching evolution was observed in six out of 10 patients during progression from SMM to symptomatic MM. This group had a relatively long time to progression (median, 23 months), which reflects the time necessary for the tumor to reach the required genetic complexity. In the remaining four patients, static progression from SMM to MM was observed, characterized by stable subclonal structure of the disease. In these cases, time to progression was relatively short (median, 5.5 months). In another study, genetic structure of SMM and MM was shown to be similar, and the changes in the subclonal structure of the tumor were observed in all the studied cases during progression from SMM to MM [29].

Dutta et al. analyzed paired samples from five patients with MGUS and five patients with SMM obtained at the diagnosis of asymptomatic plasma cell dyscrasia and after the progression of MGUS or SMM to symptomatic MM [60]. Both stages of asymptomatic plasma cell dyscrasias were characterized by a heterogeneous subclonal structure with an average of seven and eight subclones identified in patient samples from MGUS and SMM, respectively. Importantly, mutations in driver genes were found both at the clonal and the subclonal levels. There was no quantitative increase in the mutational load and only subtle changes in the subclonal architecture of the tumor, with the appearance of new progeny subclones and/or the disappearance of the ones previously observed during progression from MGUS/SMM to MM [60]. Similarly to the static progression model described in the previously cited study, this group had a relatively short time to progression to symptomatic MM (with a median of 38 and 14 months for patients with MGUS and SMM, respectively). Noteworthy, previous studies using whole-exome sequencing (WES) showed that the genetic structure of MGUS was less complex than that of symptomatic MM, and the progression of MGUS to MM was accompanied by a significant increase in the mutational load [15,65].

To summarize, these data suggest that the progression of MGUS/SMM to symptomatic MM requires formation of a specific subclonal structure of the tumor. However, the progression of MGUS/SMM to symptomatic MM is not solely due to the clonal evolution of plasma cells as the tumor microenvironment also undergoes evolutionary changes simultaneously [21,66,67]. It cannot be excluded that a specific subclonal structure of plasma cells is required for cooperation between subclones, as well as for interactions between subclones and the tumor microenvironment, which ultimately results in the acquisition of an invasive phenotype (e.g., by promoting its growth or the abolition of the so-called immune surveillance) and is clinically expressed by the progression of MGUS/SMM to symptomatic MM [21,35,60,68]. The results of the recent analysis of the cytogenetic evolution of SMM are consistent with this hypothesis, showing increased risk of progression to symptomatic MM in the cases where the high-risk dominant clone coexisted with subclones with additional cytogenetic aberrations regardless of whether these were standard- or high-risk cytogenetic aberrations [69]. Similarly, a recently published study of clonal evolution during progression from SMM to MM showed that in each case assessed, there was a significant change in the subclonal structure of the tumor in the year before progression. In 7/8 cases, the clonal evolution followed the pattern of a branching tree; in one case, linear evolution was observed [62].

## 5. Genomic Landscape of Multiple Myeloma

Genomic instability, a hallmark of all neoplastic diseases, is the source of genetic heterogeneity of MM not only at the individual patient level (intrapatient genetic heterogeneity), but also between patients with the same diagnosis (interpatient heterogeneity) [35,70,71]. Therefore, the molecular landscape of newly diagnosed MM is heterogeneous as there is no universal mutation pattern typical for MM [16,17,18,19,72,73,74]. High-throughput sequencing techniques, however, made it possible to identify over 80 driver genes that are repeatedly mutated in MM patients (Table 2) [16,17,18,19,72,73,74].

In the newly diagnosed MM, a significant percentage of mutations in driver genes is subclonal [17,18,19,20,72], indicating that these mutations develop later in the course of tumor evolution. The acquisition of mutations in driver genes during the development of MM is secondary to primary genomic events, i.e., initiators of oncogenesis in MM, namely hyperdiploidy and translocations involving the immunoglobulin heavy chain locus (IGH), including t(4; 14), t(6; 14), t(11; 14), t(14; 16), and t(14; 20) [21]. WES and WGS of more than 800 MM cases led to characterization of main processes that determine mutagenesis in myeloma cells [75]. Mutation signature analysis showed that mutagenesis of MM is the result of natural processes of aging, defects in DNA repair mechanisms, and mutagenic activity of enzymes, namely activation-induced deaminase (AID) and apolipoprotein B mRNA editing enzyme, catalytic polypeptide (APOBEC).

AID activity plays a key role in the early stage of neoplastic transformation of plasma cells [75,76]. At later stages of tumor development, mutagenesis differs depending on the primary genomic lesions. For example, mutational landscape in MM with a hyperdiploid karyotype is dominated by mutational signatures associated with aging, whereas in MM with t(4; 14), mutations related to the APOBEC activity and DNA repair defects are predominant. The latter are also typically observed in MM with t(14; 16) and t(14; 20) while MM with t(11; 14) was shown to carry signatures associated with a defect in DNA repair [75,76,77]. This observation suggests that the acquisition of mutations in some driver genes is not a random process and at least to some extent depends on the primary genomic event initiating oncogenesis in MM. For example, (1) *CDKN1B, FUBP1, NFKB2, PRDM1, PTPN11, RASA2, RFTN1,* and *SP140*, (2) *FGFR3, PRKD2,* and *ACTG1*, (3) *CCND1, IRF4, LTB,* and *HUWE1*, (4) *MAF*, (5) *MAFB* are mutated in patients with hyperdiploidy, t(4; 14), t(11; 14), t(14; 16), and t(14; 20), respectively [17].

The molecular heterogeneity of MM is expressed not only by the wide catalog of mutated driver genes, but also by the fact that the frequency of mutations in individual driver genes in relation to the entire population of patients with newly diagnosed MM is relatively low. Data from the largest population of patients with newly diagnosed MM sequenced (*n* = 1273) indicate that mutations in the most commonly mutated driver genes in MM, i.e., *KRAS, NRAS, DIS3, FAM46C*, and *BRAF* occur in 22, 17, 10, 9, and 8% of patients, respectively, while the frequency of mutations in other driver genes ranges from 0.3% to 5.5% [17]. Such distribution of mutations determines wide interpatient genetic heterogeneity and suggests that the development of MM does not follow the universal evolutionary pattern.

The genetic landscape of the newly diagnosed MM also includes biallelic inactivation (double-hit event), i.e., homozygous deletions of a specific gene or coexistence of deletion of one allele with an inactivating mutation in the other allele. Biallelic inactivation is most often observed for the following tumor suppressors: *DIS3, TRAF3, FAM46C, TP53, RB1*, *CYLD, ATM, BIRC3, TGDS, HUWE1* [17]. Recently, in addition to gene mutations and chromosomal aberrations, mutations in non-coding DNA were reported, including the promoter and cis-regulatory regions, resulting in a significant change in the expression of certain genes, including *PAX5, MYC, SP110,* and *HOXB3* [73].

## 6. Spatial Genetic Heterogeneity of Multiple Myeloma

The introduction of the positron emission tomography/computed tomography (PET/CT) in the diagnosis of MM provided evidence that MM is a spatially heterogeneous disease showing a tendency to form focal lesions in the bone marrow with different metabolic activity (patchy disease) [78,79]. The genetic structure of these lesions expressed in the local subclonal architecture of the tumor also, as in the case of many solid tumors [80], remains heterogeneous resulting in wide spatial genetic heterogeneity. Rasche et al. compared bone marrow samples from posterior superior iliac spine with samples obtained from at least one additional disease site captured by CT [12]. Paired samples differed in terms of single nucleotide variants (SNV), indels, and copy number variants (CNV), including prognostically significant cytogenetic aberrations such as del(17p), amp(1q21), and del(1p). A different mutation profile (both clonal and subclonal) in at least two separate locations of MM was observed in 75% of patients [12]. This spatial genetic heterogeneity positively correlated with the dimensions of focal lesions from which additional samples were collected [12], which supports the hypothesis of independent course of clonal evolution in different locations of the disease.

## 7. The Role of the Tumor Microenvironment in Multiple Myeloma Evolution

The role of the bone marrow microenvironment in the survival and proliferation of plasma cells and the development of treatment resistance in MM is well-established [66]. Tumor microenvironment is a complex heterogeneous structure [56,81,82] providing selection forces that enforce continuous competition between the subclones. Shen et al., using an animal model of MM to study clonal behavior within the bone marrow microenvironment, showed that distant bone marrow sites can only be colonized by a few clones well-adapted to the local microenvironment [82], again demonstrating that autonomous factors of plasma cells are not sufficient to independently drive MM progression. The complex interplay between cancer genomics and the tumor microenvironment can be better understood through single-cell technologies. Recently, Liu et al. profiled sequential samples from 14 patients at single-cell resolution and found that MM progression is driven by coevolution of tumor and immune microenvironment cells [83], emphasizing that the microenvironment is not a simple passive source of selection forces for malignant plasma cells but rather an active and dynamic cocreator of myeloma development and progression.

In conclusion, as in the microecosystem, myeloma development and progression are driven by clonal competition and clonal cooperation, which occur in the context of dynamic changes in the cellular and noncellular components of the tumor microenvironment. The complexity of these processes is further compounded by the reciprocal relationship between subclones and the microenvironment as subclones can interact with the microenvironment and modify its structure to favor the selection and expansion of subclones with specific phenotypic features [51].

## 8. Clonal Evolution of Multiple Myeloma during Therapy

Anticancer therapy forms strong selective pressure modifying the course of clonal evolution of cancer cells. However, although sensitive subclones become eradicated, anticancer treatment usually favors selection of the drug-resistant subclones initially present in the tumor mass or emerging during therapy [18,40,84,85] (Figure 3).

In a study by Corre et al. (Table 1), change in the tumor (sub)clonal structure during MM progression was observed in most patients. The emergence of treatment resistance was possibly due to selection of preexisting subclones or acquisition of new subclonal mutations during antimyeloma treatment or during further MM evolution at the stage of minimal residual disease (MRD). In one third of patients the mutation profile remained unchanged, but the appearance of new CNV (predominantly amp(1q21) and del(1p)) was reported. Although the patients received homogeneous treatment, no specific mutation profile was observed to be selected during therapy, suggesting that genetically distinct tumors progress following different evolutionary patterns [18]. Consistent findings were reported in a similar study using targeted sequencing to analyze the clonal evolution of MM during antimyeloma treatment [74].

Jones et al. (Table 1) performed WES in sequential samples from the participants of the randomized phase 3 Myeloma XI trial comparing thalidomide with lenalidomide at induction and lenalidomide maintenance with no maintenance in both transplant-eligible and transplant-ineligible patients [63]. Only patients who relapsed within 30 months of the second randomization (lenalidomide vs. observation arm) were included in the analysis. Relapse was associated with a change in the genetic structure of the disease (both at the SNV and CNV levels). The branching evolution model, found in 2/3 of the patients, was the dominant pattern reported. In every fifth patient, clonal evolution followed a linear evolution model, while in only seven patients (12.5%) a stable subclonal structure of the tumor was observed during MM progression. The key findings from this study indicate that the pattern of clonal evolution is related to the response to antimyeloma therapy. The model of a stable subclonal structure was observed only in the patients who did not achieve CR. On the contrary, in the patients who achieved CR, and specifically in those with MRD eradication, clonal evolution followed either the branching or the linear model. Importantly, lenalidomide maintenance did not influence the pattern of MM evolution [63]. It should be highlighted, however, that the stable subclonal structure observed in this study may have resulted from sampling bias as genomic analysis of a single sample gives limited insight into the actual genetic structure of a spatially heterogeneous tumor [12,68].

The models mentioned above are not the only patterns of MM evolution during IMiD-based therapy. Johnson et al. (Table 1) [64] showed that in nearly every fifth patient exposed to IMiDs, the clonal dynamics followed the model of neutral evolution, which means that in these cases the selection pressure did not play a significant role in shaping the subclonal architecture of the tumor. This pattern of evolution was found to be associated with the presence of IGH translocations as the primary oncogenic event and correlated with poor prognosis [64].

Weinhold et al. (Table 1) using WES gave a broad insight into the spectrum of changes in the MM molecular landscape between diagnosis and relapse [14]. Progression was associated with a significant increase in mutational load, including mutations in the *TP53, SETD2, PRDM1, NRAS, KRAS, HDAC4, FANCA, FAM46C, EGFR, DOT1L, CDKN2C*, and *BRAF* genes, acquisition of new CNV, including del(17p), amp(1q21), del(1p), del(6q), and del(16q), as well as the acquisition of new translocations including the *MYC* gene. *TP53* mutations (SNV, indels) were present in 45% of the patients at relapse while double-hit events involving the *TP53* gene were observed in nearly every third patient in this group. Biallelic inactivation of other tumor suppressor genes (including *CDKN2C, FAM46C, PTEN, BIRC2, RB1, TRAF3, CYLD,* and *WWOX*) occurred in every second patient [14]. In most cases, evolution followed a branching pattern. However, in 1/3 of the patients, other evolutionary patterns were observed, including linear evolution, differential clonal response (a pattern in which there was only a change in the proportion between subclones), genetic drift, and a stable subclonal structure. Interestingly, the stable subclonal structure at MM progression was associated with the primary refractory disease.

Patients with double-refractory MM (DR-MM), i.e., resistant to both IMiDs and PIs, have unfavorable prognosis, with the median overall survival of 8 months [86]. As a recent WES study has shown, these MM cases are characterized by a significant subclonal heterogeneity, with the number of subclonal mutations exceeding the number of clonal mutations in every third patient [59]. Although in this study paired samples were not compared, DR-MM had a significantly more complex subclonal architecture than newly diagnosed MM. While the TP53 pathway was mostly mutated (45% of the cases), the genes potentially involved in resistance to IMiDs (*CRBN, DDB1, RBX1, CUL4B*) and PIs (*PSMB5, PSMB8, PSMB9, PSMD1, PSMG2, XBP1*) were rarely affected (32.5% and 10% of the cases, respectively), suggesting that the acquisition of mutations in these genes is not a key mechanism in the emergence of resistance to IMIDs- or PIs-based therapy. Despite clinical and prognostic homogeneity of the studied group, it was still not possible to identify a universal genetic pattern that would determine the emergence of treatment resistance [59].

The presence of MRD after treatment is an unfavorable prognostic factor in MM [87,88]. The MRD is a reservoir of drug-resistant subclones and, from the evolutionary point of view, represents the founder population for clinically overt MM relapse (Figure 3). Thus, accurate genotypic and phenotypic characterization of MRD appears to be the main step in deciphering the mechanisms of resistance to antimyeloma therapy. Recently, Paiva et al. conducted a simultaneous cytogenetic, immunophenotypic, and transcriptomic (gene expression profiling) analysis of MRD in MM patients not eligible for intensive therapy [89]. In most cases (9/12), the genetic structure of MRD differed from diagnostic samples, with the median of four different CNV per set of paired samples. Immunophenotypically, MRD subclones overexpressed integrins, chemokine receptors, adhesion molecules, and molecules involved in the interaction with dendritic cells, which once again underlines the importance of interaction with the bone marrow microenvironment in the selection of treatment-resistant subclones.

As malignancies act as microecosystems, the impact of anticancer treatment on clonal evolution should be considered in a broader perspective. In addition to direct effects on cancer cells, anticancer treatment also influences function and structure of the tumor microenvironment and modifies the spectrum of interactions between subclones and the microenvironment [31]. Similarly, therapy-induced changes in the subclonal architecture of cancer affect both (sub)clonal competition and (sub)clonal cooperation [68,80].

The emergence of drug resistance is a major problem in MM management. In the context of the structural and functional complexity of MM, it seems likely that resistance to treatment is mediated by the overall oligo (sub)clonal structure of the tumor formed during clonal evolution, consisting of subclones with different genotypes and phenotypes able to cooperate with each other and with the local microenvironment.

## 9. Methodological Limitations of Clonal Evolution Studies in Multiple Myeloma

WGS allows for the full identification of genetic lesions in the tested sample, but its wide application is limited by high sequencing costs, difficult and time-consuming result analysis, and limited possibility of interpretation of the sequencing data, especially in terms of non-coding regions of DNA [90]. In most WGS studies, the sequencing depth is relatively low (30–50×), making it impossible to identify rare genetic variants, resulting in the underestimation of the subclonal structure and genetic heterogeneity of cancer [91]. However, as the cost of sequencing continues to decline, bioinformatics tools are optimized, and the understanding of the role of non-coding DNA in oncogenesis improves, it is expected that in the future WGS will be widely used in tracking the clonal evolution, possibly also in everyday clinical practice [92]. In WES, on the other hand, the average sequencing depth is higher (100× on average), which makes it possible to identify rarer genetic variants (minor subclones). However, the analysis covers only 2% of the cancer genome (coding DNA) and the possibility to detect structural variants and CNV is limited [93]. In targeted sequencing, the average depth of sequencing is usually between 200 and 1000×, which gives a chance to detect small subclonal populations; however, tracking genetic changes during clonal evolution is possible only within an arbitrarily determined panel of genes/DNA regions [94].

In addition to these methodological limitations, genetic analysis of a single sample taken from a single tumor site makes it impossible to determine spatial genetic heterogeneity, both within a single location and between distant tumor locations (so-called “sampling bias”) [12,32,68,69]. Therefore, multiple sampling of multiple tumor sites is necessary to determine the true spatial genetic heterogeneity [12,33].

Rapid advances in molecular biology techniques, including single-cell and multi-omics sequencing, have significantly improved our ability to study spatial genetic heterogeneity and overcome the limitation of bulk sequencing mentioned above. For example, single cell RNA sequencing (scRNAseq) has been applied to study tumor heterogeneity dynamics in the course of MM progression. The results showed high interindividual variability at the single cell level, including emergence of rare malignant clones at premalignant stages, that may have important implications for personalized therapies [95]. More recently, the scRNAseq methodology has been applied in a clinical trial to study the dynamics of MM resistance mechanisms in highly resistant MM patients. Interestingly, the study revealed new, potentially druggable resistance mechanisms and therefore opened new possibilities for clinical utilization of modern advances in high-resolution sequencing technologies [96].

## 10. Clinical and Diagnostic Implications of Clonal Evolution in Multiple Myeloma

In the light of the current definition of symptomatic MM, based on the morphological criteria (presence of at least 10% of clonal plasmocytes in the bone marrow) and the clinical and laboratory-radiological criteria (so-called “SLIM-CRAB” criteria) [26], the existence of genetic heterogeneity does not significantly affect the diagnosis of symptomatic MM. However, macrofocal MM, with heterogenous bone marrow involvement, may pose a diagnostic challenge, as it often fails to meet the morphological diagnostic criterion of symptomatic MM if bone marrow biopsy is obtained from the iliac spine only [12,78,97].

In the course of clonal evolution, the proportion of subclones secreting different monoclonal proteins may change during treatment. In consequence, the clinical phenotype of the disease may change, and MM primarily secreting complete immunoglobulin may transform into non-secretory MM [98] or light chain MM (so-called “light chain escape”) [99,100]. Change of the original monoclonal component observed in the course of clonal evolution of MM may significantly interfere with the assessment of treatment response and post-treatment follow-up; in particular, it may result in the failure to capture disease progression in a timely manner.

It is expected that in the nearest future, MRD monitoring will become the standard of care in the treatment of patients with MM. The so-called phenotypic shift resulting from the clonal evolution of the disease during or after treatment limits the ability to monitor MRD using polymerase chain reaction (PCR)-based methods but has no major effect on the evaluation of MRD by flow cytometry and NGS techniques [101,102].

## 11. Cytogenetic Risk Stratification in the Shade of Multiple Myeloma Evolution

The prognostic significance of baseline cytogenetic aberrations in newly diagnosed MM is well-documented [103,104] and cytogenetic risk stratification of MM patients is widely used in clinical practice [105]. Cytogenetic aberrations have been shown to have a significantly greater prognostic impact in MM than mutations in specific genes [72] and there is increasing evidence that the evolution of cytogenetic aberrations over time, and in particular the acquisition of high-risk aberrations (e.g., del(17p)), has an adverse effect on MM patient prognosis [106,107,108]. Therefore, we believe that cytogenetic assessment should be repeated with each subsequent MM relapse. The spatial heterogeneity of MM significantly limits thorough assessment of cytogenetic risk. Considering the data presented by Rasche et al. in two out of the six patients (33%) with del(17p), this aberration was found in specific sites only. Similarly, the presence of chromosome 1 aberrations (del(1p) or amp(1q21)) showed spatial differentiation in every fifth patient [12]. This means that in a significant percentage of MM cases, prognostically significant cytogenetic aberrations may be missed due to the spatial genetic heterogeneity.

## 12. Tumor Evolution as a Target for Anticancer Therapy

Although therapeutic strategies in MM are improving year by year, they are not adapted to the inter- and intrapatient genetic heterogeneity. As tumors change over time, it is now believed that to achieve long-term disease control, treatment protocols should take the evolutionary dynamics of cancer into account [31]. The aim of dynamic therapy, represented by two basic models (i.e., sequential therapy and adaptive therapy), is to adapt anticancer treatment to the evolutionary processes that drive tumor progression. In the sequential therapy, successive anticancer drugs with the ability to overcome resistance of the subclones selected earlier during therapy are introduced into the treatment [35]. This is an important alternative to the so-called “all-in-one strategy” in which anticancer agents with different mechanisms of action are given simultaneously within a single treatment regimen. The model of adaptive therapy, in turn, targets relationships between the subclones to achieve a therapeutic effect [109]. It is assumed that the dominant subclone, which is sensitive to the treatment, inhibits the expansion of the resistant subclone. The aim of adaptive therapy is to reduce the treatment-sensitive subclonal population only to a level at which its suppressive effect on resistant subclones is still maintained.

Dynamic therapy is a truly personalized anticancer treatment. However, its introduction into clinical practice requires significant progress in understanding the processes that drive clonal evolution of cancer. Bone marrow aspiration biopsy is not a suitable method to track the clonal changes that occur during MM progression, mainly due to its invasive nature limiting the possibility of serial sampling at short intervals. Moreover, as discussed earlier, genetic analysis of plasma cells isolated from a single bone marrow aspirate does not provide accurate insight into the overall subclonal complexity of MM (sampling bias). Noninvasive tracking of clonal evolution using cell-free DNA (cfDNA) or DNA isolated from circulating tumor cells (so-called “liquid biopsy”) allows overcoming these limitations. Manier et al. recently compared the clonal and subclonal structures of MM using WES on the DNA extracted from CTC, cfDNA, and from plasma cells isolated from a single bone marrow aspirate [110]. Almost all clonal mutations (99%) detected in the bone marrow aspirate were also detected in CTC and cfDNA. Subclonal mutations, in most cases, were detected in both CTC and cfDNA, as well as in bone marrow, but in a few cases some subclonal mutations were detected only in CTC and cfDNA [110]. Thus, the subclonal structure of MM determined using liquid biopsy was more complex than that estimated from a single bone marrow aspirate, confirming the usefulness of liquid biopsy in monitoring clonal evolution in the clinical setting.

## 13. Conclusions

Over the last few years, a growing number of high-throughput sequencing studies have captured several evolutionary patterns involved in the development and progression of MM. These processes are deeply involved in shaping the clinical course of the disease, influencing treatment response and patients’ survival. Although today MM patients do not benefit from extensive genetic evaluations and estimation of tumor evolutionary patterns yet, thorough understanding of the biological processes underlying tumor heterogeneity and behavior under therapeutic pressure may result in improved risk stratification and progression definition as well as development of truly personalized therapy for myeloma in the near future. Dynamic therapy or clinically available noninvasive tracking of clonal evolution using cfDNA are just a few examples of clinical utilization of the growing body of knowledge on tumor evolution making personalized care in MM possible.

## Figures and Tables

**Figure 1 diagnostics-11-01534-f001:**
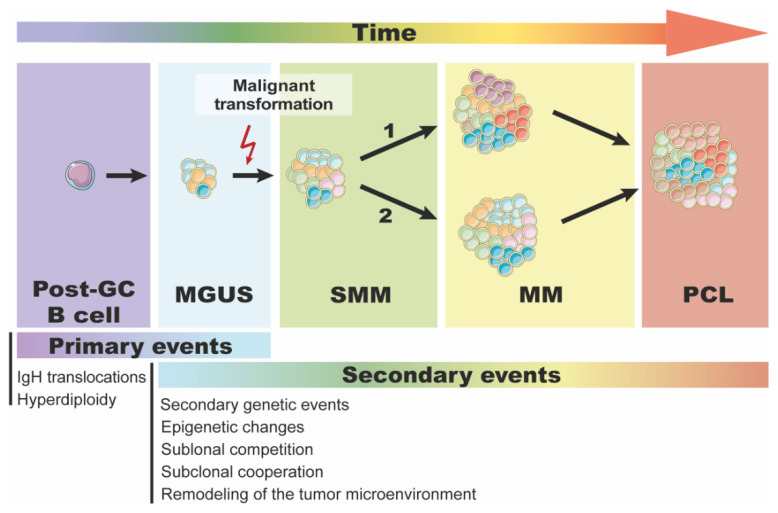
Evolution of plasma cell dyscrasias. Both primary and secondary genetic events are required for malignant transformation of a post-GC B cell to MM. New genetic events accumulate in malignant plasmocytes over time affecting the fitness of subclones and leading to further expansion of some and to the extinction of others. Progression from MGUS/SMM to MM is driven by subclonal competition (and likely by its cooperation) and by simultaneous changes in the tumor microenvironment. (1) Branching clonal evolution—the dominant evolutionary pattern in the progression of SMM to MM, with some subclones disappearing (pink, light blue) and others appearing (red, purple, brown) over time; (2) static clonal progression—evolutionary pattern in the progression of SMM to MM, in which the tumor’s subclonal architecture does not change significantly over time. Abbreviations: GC, germinal center; IgH, immunoglobulin heavy chain; MGUS, monoclonal gammopathy of undetermined significance; MM, multiple myeloma; SMM, smoldering multiple myeloma; PCL, plasma cell leukemia. The figure was prepared using images provided by Servier Medical Art (https://smart.servier.com accessed on 23 August 2021).

**Figure 2 diagnostics-11-01534-f002:**
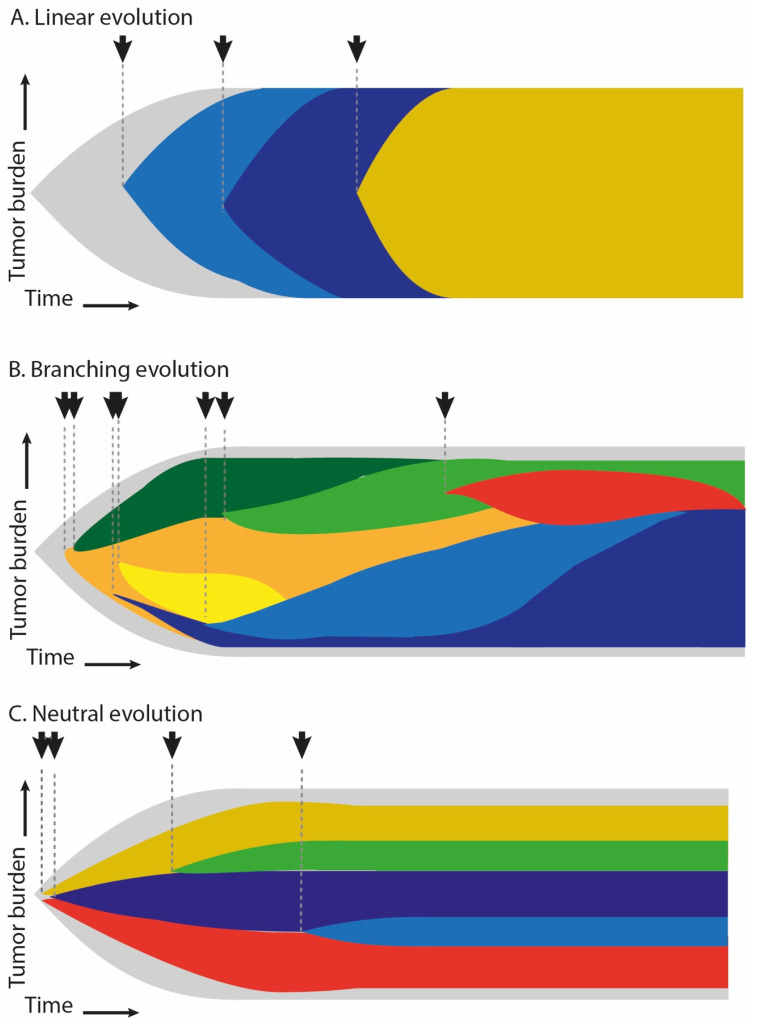
Main models of clonal evolution in multiple myeloma. Each color represents a single subclone. Progeny subclones arising during evolution are marked with arrows.

**Figure 3 diagnostics-11-01534-f003:**
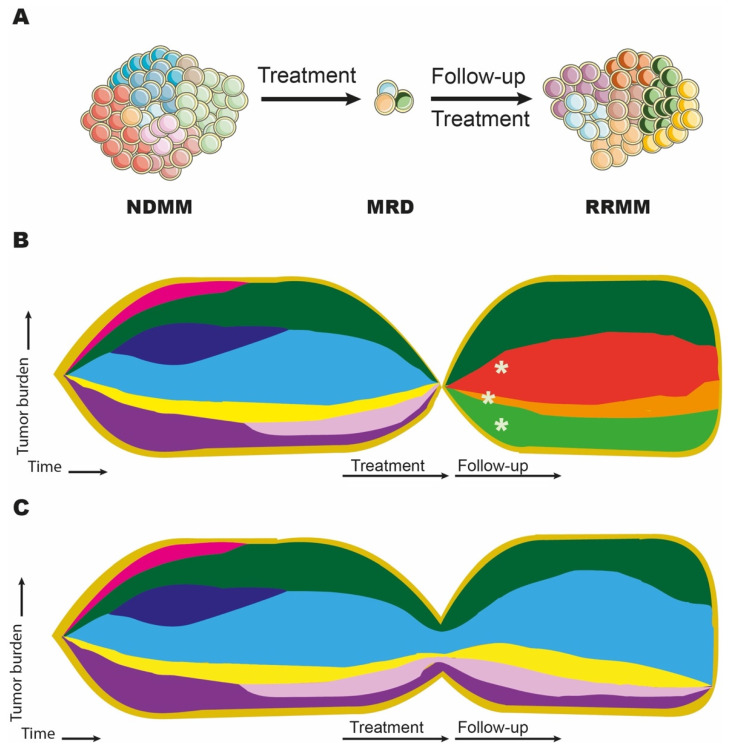
Clonal evolution of multiple myeloma during therapy. Each color represents a single subclone. (**A**) Treatment-resistant subclones evolve at the MRD stage, being the source for overt disease relapse; (**B**) Selection of preexisting subclones and formation of new subclones (marked with stars) during antimyeloma therapy. (**C**) Change in the subclonal architecture of multiple myeloma during therapy with the selection of some subclones (green, light blue, yellow, purple, lavender) and the disappearance of others (pink, dark blue). Abbreviations: MRD, measurable residual disease; NDMM, newly diagnosed multiple myeloma; RRMM, relapsed/refractory multiple myeloma. The figure was prepared using images provided by Servier Medical Art (https://smart.servier.com accessed on 23 August 2021).

**Table 1 diagnostics-11-01534-t001:** Selected studies using next-generation sequencing to track clonal evolution in MGUS, SMM, and MM. Abbreviations: MGUS, monoclonal gammopathy of undetermined significance; SMM, smoldering multiple myeloma; MM, multiple myeloma; WES, whole-exome sequencing; WGS, whole-genome sequencing; Dx, diagnosis; VTD, bortezomib, thalidomide, dexamethasone; autoHSCT, autologous hematopoietic stem cell transplantation; NDMM, newly diagnosed MM; RRMM, relapsed/refractory MM; OS, overall survival.

**MGUS/SMM**				
**Studied Population**	**Type of Analysis**	**Treatment**	**Major Findings**	**Reference**
MGUS–MM and SMM–MM	WES	no	Subclones identified at MM were already present at the MGUS/SMM stage. Disease progression from MGUS/SMM to MM was mainly characterized by clonal stability as only subtle changes occurred in the subclonal tumor architecture during progression.	Dutta et al. [60]
SMM–MM	WGS	no	Two patterns of progression from SMM to MM were observed: a static progression model and branching clonal evolution.	Bolli et al. [61]
SMM–MM	Targeted and WES	no	The tumor subclonal complexity increased at least one year before progression from SMM to MM and branching clonal evolution was the dominant pattern of progression.	Boyle et al. [62]
SMM–MM	Targeted and WES	no	SMM was found to be a genetically mature entity; however, in all cases of progression from SMM to MM, clonal evolution with subclonal cancer fractions changing over time was detected.	Bustoros et al. [29]
**MM**				
**Studied Population**	**Type of Analysis**	**Treatment**	**Major Findings**	**Reference**
Dx and the first relapse	Targeted sequencing (246 genes)	4× VTD, autoHSCT, 2× VTD	Different evolutionary patterns of progression from NDMM to RRMM were observed, including selection of very rare subclones present at diagnosis, appearance or disappearance of mutations, and genetic stability. Drug resistance could arise through acquisition of new mutations in the driver genes or selection of preexisting (sub)clonal mutations.	Corre et al. [18]
Dx and the first relapse	WES	Myeloma XI trial protocol; maintenance with lenalidomide vs. observation	Different evolutionary patterns of progression from NDMM to RRMM were observed, including branching clonal evolution (2/3 of patients), linear evolution, and stable subclonal structure. Evolutionary pattern appeared to be related to the depth of treatment response.	Jones et al. [63]
Dx and the first relapse	WES	Myeloma XI trial protocol	Neutral evolutionary dynamics observed in 17–20% of patients during progression from NDMM to RRMM was shown to have a negative impact on OS.	Johnson et al. [64]
Dx and the first relapse	WES	Total therapy protocols	Different evolutionary patterns of progression from NDMM to RRMM were observed, including branching clonal evolution (predominant), linear evolution, neutral evolution, and stable subclonal structure.	Weinhold et al. [14]

**Table 2 diagnostics-11-01534-t002:** The most frequently mutated genes in multiple myeloma (MM) patients.

Biological Process/Pathway	Mutated Enes
MAPK/ERK pathway	*KRAS, NRAS, BRAF, PTPN11, RASA2, NF1, PRKD2, FGFR3, DUSP2, LEMD2*
NF-kB pathway	*TRAF2, TRAF3, CYLD, NFKB2, NFKBIA, MAP3K14*
PI(3)K-AKT/mTOR pathway	*PIK3CA, TCL1A*
Cell cycle	*CCND1, CCND2 RB1, CDKN2C, CDKN1B, BTG1*
Transcription factors	*EGR1, MAF, MAFB, MAML2, MAX, NFKB2, XBP1, ZFP36L1, ZNF292, SP140, FUBP1, PIM1, DTX1, BHLHE41, RPRD1B*
RNA metabolism	*DIS3, FAM46C, SF3B1, PABPC1, RPL5, RPL10, RPS3A*
Epigenetics	*HIST1H1B, HIST1H1D, HIST1H1E, HIST1H2BK, HIST1H4H, KMT2C, CREBBP, ARID1A, KMT2B, ATRX, EP300, SETD2, TET2, KDM5C, ARID2, DNMT3A, KDM6A, NCOR1, IDH1, IDH2, BCL7A*
DNA repair	*TP53, ATM, ATR, BRCA2*
Immune response	*ABCF1, KLHL6, LTB, RFTN1, SAMHD1, PRDM1, IRF1, IRF4, CD96*
Apoptosis	*BAX*
Telomere maintenance	*POT1*
Ubiquitin–proteasome pathway	*UBR5, HUWE1, FBXW7, RPN1*
Cell metabolism	*TGDS, MAN2C1, FTL, SGPP1*

Abbreviations: MAPK/ERK, mitogen-activated protein kinases/extracellular signal-regulated kinases; NF-kB, nuclear factor kappa B; PI(3)K, phosphoinositide 3-kinase; AKT, Ak strain transforming; mTOR, mammalian target of rapamycin.

## Data Availability

Not applicable.

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
