# Peer review of "Clonal Evolution of Multiple Myeloma—Clinical and Diagnostic Implications"

_diagnostics, 2021, doi:10.3390/diagnostics11091534_

Round 1
Reviewer 1 Report
The manuscript presented by the authors has reviewed the recent advances of clonal evolution patterns in MM from different aspects, and also discussed the clinical implications from genomic aberrations. The paper was well organized and included the most updated data and findings related. They were well displayed and informative. I have nothing to add in.
Author Response
The authors are thankful for the kind comments from the reviewer. In the reviewing process, slight corrections to the introduction section have been made. Otherwise, the manuscript has not been changed.
Reviewer 2 Report
Very well-written and comprehensive review on the Subject, presenting the different modes of evolution and spatiotemporal dynamics particularly well. I would propose a few minor changes.
- An extra paragraph in the introduction presenting the new horizons clearly opened be the rapid advancements in NGS technologies.
- A bit more extensive mention of single-cell NGS, particularly scRNAseq, which have achieved breakthrough fundamental discoveries regarding new potential therapeutic directions (for instance: Cohen, Y.C., Zada, M., Wang, SY. et al., Nat Med 27, 491–503, 2021).
Author Response
Dear Reviewer,
Please find below the responses to your comments.
Comment #1
An entirely new paragraph on the advances in NGS technologies has been added to the Introduction section. In addition, some minor editing has been applied to this section as well to make it more readable (lines 31-66).
Comment #2
As suggested, a new paragraph on the advances in single-cell sequencing techniques and their input into our understanding of myeloma biology has been added (lines 498-508) together with 2 references (ref#95 and ref#96).